# Mathematical modeling suggests heterogeneous replication of *Mycobacterium tuberculosis* in rabbits

**Vitaly V. Ganusov**[1]*, **Afsal Kolloli**[2], **Selvakumar Subbian**[2¤]*

**1** Host-Pathogen Interactions program, Texas Biomedical Research Institute, San Antonio, Texas, United States of America, **2** Public Health Research Institute, The State University of New Jersey, Newark, New Jersey, United States of America

¤ Current address: Public Health Research Institute, New Jersey Medical School, Rutgers, The State University of New Jersey, Newark, New Jersey, United States of America
* vitaly.ganusov@gmail.com (VVG); subbiase@njms.rutgers.edu (SS)

## Abstract

Tuberculosis (**TB**), the disease caused by *Mycobacterium tuberculosis* (**Mtb**), remains a major health problem with 10.6 million cases of the disease and 1.6 million deaths in 2021. It is well understood that pulmonary TB is due to Mtb growth in the lung but quantitative estimates of rates of Mtb replication and death in lungs of patients or animals such as monkeys or rabbits remain largely unknown. We performed experiments with rabbits infected with a novel, virulent clinical Mtb isolate of the Beijing lineage, HN878, carrying an unstable plasmid pBP10. In our in vitro experiments we found that pBP10 is more stable in HN878 strain than in a more commonly used laboratory-adapted Mtb strain H37Rv (the segregation coefficient being $s = 0.10$ in HN878 vs. $s = 0.18$ in H37Rv). Interestingly, the kinetics of plasmid-bearing bacteria in lungs of Mtb-infected rabbits did not follow an expected monotonic decline; the percent of plasmid-bearing cells increased between 28 and 56 days post-infection and remained stable between 84 and 112 days post-infection despite a large increase in bacterial numbers in the lung at late time points. Mathematical modeling suggested that such a non-monotonic change in the percent of plasmid-bearing cells can be explained if the lung Mtb population consists of several (at least 2) sub-populations with different replication/death kinetics: one major population expanding early and being controlled/eliminated, while another, a smaller population expanding at later times causing a counterintuitive increase in the percent of plasmid-bearing cells. Importantly, a model with one kinetically homogeneous Mtb population could not explain the data including when the model was run stochastically. Given that in rabbits HN878 strain forms well circumscribed granulomas, our results suggest independent bacterial dynamics in subsets of such granulomas. Our model predictions can be tested in future experiments in which HN878-pBP10 dynamics in individual granulomas is followed over time. Taken together, our new data and mathematical modeling-based analyses illustrate differences in Mtb dynamics in mice and rabbits confirming a perhaps somewhat obvious observation that "rabbits are not mice".

**Data Availability Statement:** The data from the paper along with the codes are available on github: https://github.com/vganusov/rabbits_replication_clock/tree/main. Code sources: All analyses have been primarily performed in Mathematica (ver 12).

Stochastic (Gillespie) simulations were done in R using GillespieSSA2 library.

**Funding:** This work was supported in part by the NIH/NIAID grants R01AI158963 to VVG and R01AI127844 to SS. The funders had no role in study design, data collection and analysis, decision to publish, or preparation of the manuscript.

**Competing interests:** The authors have declared that no competing interests exist.

## Author summary

How quickly Mycobacterium tuberculosis (**Mtb**) replicates and dies in lungs of infected individuals is likely to determine the outcome of the infection—either control/clearance of the bacteria by the host immune response or progression to active disease, tuberculosis (**TB**). And yet, only a few studies, primarily in mice, rigorously estimated the rates of Mtb replication and death during an in vivo infection. We infected rabbits with a novel clinical isolate of Mtb carrying an unstable, "replication clock" plasmid and followed the dynamics of bacteria over time. Interestingly, previous methods developed to estimate Mtb replication and death rates using similar data for Mtb infection of mice failed to describe our novel data on Mtb dynamics in rabbits; we showed that heterogeneous dynamics of Mtb in semi-independent subpopulations in lungs of Mtb-infected rabbits may be one explanation of this failure of the method. Our results highlight potential differences in Mtb dynamics in different mammalian hosts and suggest ways to evaluate heterogeneity of Mtb replication and death rates in vivo.

## Introduction

Tuberculosis (**TB**), a disease caused by *Mycobacterium tuberculosis* (**Mtb**), remains a global health threat. In 2021, 1.6 million people died due to TB despite available effective chemotherapy [1]. The infection with Mtb results in heterogeneity of disease outcomes such as bacterial clearance, bacterial persistence, and active disease with and without lung cavitation [2–4]. Several studies suggested that between quarter to a third of the world's population has evidence of Mtb infection [5]. While most are not at risk to develop active disease, individuals who do progress to TB seem to do so within 1–2 years after exposure [6, 7].

Studying TB progression in humans is difficult as most exposed individuals do not develop the disease, and thus, it requires dedicated long-term cohort studies (e.g., [6, 8]). Most of what we know about within-host Mtb dynamics early after infection comes from animal studies [9–12]. Yet, despite decades of research, the pathophysiology of TB and the replication dynamics of Mtb during acute and chronic stages of infection remains poorly understood. Earlier studies reported that Mtb can persist in non-replicating status in tissues especially under hypoxic conditions, which may help it resist antibiotic-mediated killing [13]. Studies in B6 mice, infected via aerosol with a conventional dose of Mtb (about 100 colony-forming units, **CFU**), have shown that bacterial numbers in the lung increase during acute phase of infection but become almost static during the chronic phase of infection [14–17]. This condition was defined as "static equilibrium", where the bacteria remain viable but do not divide [14]. Another study reported that there is no marked difference in the number of CFU and the number of chromosomal equivalents (**CEQs**) supporting the hypothesis of non-replicative persistence of Mtb during chronic phase of infection [18]. In contrast to these results, other studies conducted in zebrafish and mice reported that Mtb enters in a dynamic life status and replicates in the host during chronic stage of infection [19, 20]. In addition, isoniazid has been successfully used to treat Mtb infection of humans or mice [18, 21]. Because isoniazid is effective only against actively replicating bacteria, success of isoniazid-based treatment suggests that during Mtb infection bacteria are not dormant but do replicate, and apparent constancy of bacterial numbers in latency/chronic infection is due to a delicate balance between Mtb replication and death rates. This balance is most likely maintained by immune response to Mtb [17].

Gill *et al.* [17] elegantly addressed this question of static equilibrium in a mouse model of TB. They used H37Rv strain of Mtb transformed with an unstable plasmid pBP10, dubbed a "replication clock plasmid". The plasmid was lost at a constant rate with each cell division in vitro allowing to count the number divisions cells have undergone over time. By combining the data on the total number of bacteria and percent of plasmid-bearing cells in the population with mathematical modeling, they estimated the rate of Mtb division and death during the first 110 days of Mtb infection in lungs of B6 mice. They found that Mtb replicates at substantial rates during the chronic phase of infection—about 4 fold lower than in acute phase of infection—thus, challenging the conventional concept that Mtb enters dormancy during chronic infection of mice [17]. We have recently shown how the plasmid segregation probability influences estimates of Mtb replication and death rates, and quantified the fraction of non-replicating bacteria that may be formed in these experiments [22]. More recent work used the ratio of abundance of short- vs. long-lived ribosomal RNA (RS ratio) to further confirm that Mtb may indeed be replicating during the chronic stage of infection in murine lungs [23].

While infection of B6 mice with Mtb is a widely used animal model of TB it has some limitations. Typically mice are infected with conventional doses ($\sim$ 100 bacteria) that likely exceed infection dose of humans [24, 25]. At such high doses given a small size of murine lung, nearly all areas of the lung have sign of inflammation, but this is not typically observed in humans with pulmonary TB [6]. In contrast, rabbits or monkeys exposed to similar or lower Mtb doses do develop localized lesions that may progress to well-circumscribed granulomas, in part, due to a larger size of the lungs [26–29]. Interestingly, the number of bacteria recovered from individual lesions/granulomas from the same animal can vary orders of magnitude but whether such variability arises due to differences in replication rate or death rate (or both) of bacteria in different lesions remains largely unknown [27, 30, 31]. Furthermore, in the same animal some lesions may heal and some may become more active (determined, for example, by using PET/CT technology, [30]) but the reasons for such discordant dynamics (e.g., is this due to changes in replication and/or death rates?) remain unknown.

In this study we used a clinical Mtb isolate HN878, transformed with the replication clock plasmid pBP10, to infect rabbits via aerosol challenge. We followed the total number of bacteria and the percent of plasmid-bearing cells in rabbit lungs over time. Interestingly, the percent of plasmid-bearing cells did not decline monotonically as has been previously observed with H37Rv-pPB10 strain in B6 mice, and our previous mathematical model failed to accurately fit the observed data on Mtb CFU in the lung. Rather, a model in which Mtb population in the lung consists of at least 2 somewhat independent populations fitted the data better. Stochastic simulations of Mtb dynamics suggest that it is possible to use replication clock plasmid (and HN878-pBP10 specifically) to estimate the rates of Mtb turnover in individual lesions in lungs of rabbits, paving the way to rigorously understand why individual granulomas in a given animal vary dramatically in the number of viable bacteria.

## Materials and methods

### Ethics statement

Specific pathogen-free, female New Zealand White rabbits of 2.3 to 2.6 kg body weight were purchased from Covance (Envigo, Indianapolis, IN, USA). Each rabbit was individually housed with food and water consumption ad libitum. Animals were handled humanely according to the Association for Assessment and Accreditation of Laboratory Animal Care (AAALAC) and the United States Department of Agriculture (USDA) guidelines. All animal procedures involving Mtb, including infection and necropsy were performed in Biosafety Level-3 (BSL-3) facilities according to the protocols approved by the Institutional Animal Care

and Use Committee (IACUC) of the Rutgers University. The rabbit Mtb studies were approved by the Rutgers IACUC (protocol #12039).

## Data

**In vitro experiments.** We used a clinical Mtb strain HN878, carrying pBP10 plasmid. The plasmid was transformed into HN878 by D. Sherman's group (University of Washington) using a methodology described previously [17]. To estimate how unstable the plasmid is in this specific Mtb strain, we performed experiments with HN878-pBP10 strain grown in vitro in different conditions. Experiments were performed as described by Gill *et al*. [17] with some modifications. Specifically, Mtb HN878-pBP10 was grown in vitro starting with $2 \times 10^7$ cell/ mL in 10 mL for 3 days in 2 different media: complete Middlebrook 7H9 media ("7H9"), 1:3 7H9:PBS diluted media ("1:3 7H9") to simulate different replication rates of the bacteria. After 3 days, 1 mL of the 10mL culture was inoculated into 9 mL fresh media. The transfer was repeated 7 times. The experiment was performed with 3 independent cultures for each medium.

**In vivo experiments.** We have previously described a protocol of a rabbit model of pulmonary TB, using the clinical Mtb isolates such as HN878 or CDC1551 to infect animals via the respiratory route [32–34]. In the present study, rabbits were infected with about 500 colony forming units (CFU) of HN878 or HN878-pBP10 by aerosol. Specific pathogen-free, female, New Zealand white rabbits (*Oryctolagus cuniculus*), weighing 2.2 to 2.6 kg, were used ($n = 23$) for aerosol infection by Mtb HN878 ($n = 12$) or Mtb HN878-pBP10 ($n = 11$) in four separate experiments ($n = 2$ to 4 per time point per experiment) because our infection chamber could only hold 6 rabbits at a time. Rabbits were exposed to Mtb-containing aerosol using a nose only delivery system. At 3 hours after exposure, a group ($n = 4$) of rabbits was euthanized, and serial dilutions of the lung homogenates were cultured on Middlebrook 7H11 (Difco BD, Franklin Lakes, NJ) agar plates to enumerate the number of initial (time 0) bacterial CFU implanted in the lungs. At 28, 56, 84, and 112 days (or 4, 8, 12, and 16 weeks) post infection (**p. i.**), groups of rabbits ($n = 2$ to 4) were euthanized and left and right lungs were harvested for CFU assay by plating lung homogenates on 7H10 plates without and with kanamycin. Plasmid-bearing Mtb can grow on kanamycin-containing plates while the plasmid-free cells cannot [17].

**Determining CFU in experiments.** The standard procedure for growing Mtb, making stock and working cultures for *in vitro* and *in vivo* experiments are well-established and routinely used in various studies in the Subbian lab [32–34]. The bacterial growth media (both liquid and agar-based) were prepared in bulk quantities by the researchers in the lab (e.g, 2 liters of media and 100 plates in one preparation). The CFU plate assay conditions were maintained at same parameters through electronic regulators (i.e, 37˚C with 5% $CO_2$) in the incubators. The *in vitro* and *in vivo* plating experiments were performed by two experienced researchers using standard protocols. Therefore, the kinetics of bacterial load (with or without plasmid) difference across different time points were highly unlikely to be due to experimental variables.

## Mathematical models

There are two alternative methods of estimating rates of Mtb replication and death from the experimental data involving growth of bacteria bearing the replication clock plasmid [22]. One method proposed by Gill *et al*. [17] ("linear regressions") involves the use of linear regressions of the log-transformed numbers of bacteria found in the lungs at different times after the infection, and another method proposed by McDaniel *et al*. [22] is to fit the ordinary differential

equation-based (**ODE**) mathematical model to whole dataset assuming that model parameters are time-dependent ("fitting ODE model to data").

**Linear regressions.** In the linear regression analysis of Gill *et al.* [17], for every time interval *i* in the data (e.g., in S2 Fig the first time interval is 0–28 days post infection), three slopes can be calculated: *slopeN$_i$* (the rate of exponential change in the total number of bacteria in the time interval *i*), *slopeP$_i$* (the rate of exponential change in the number of plasmid-bearing bacteria), and *slopeF$_i$* (the rate of exponential decline in the fraction of plasmid-bearing bacteria in the population). By using two of these slopes, the rate of Mtb replication $r_i$ and death $\delta_i$ in the $i^{th}$ interval can be estimated as

$$r_i \quad = \quad SlopeF_i/s, \tag{1}$$

$$\delta_i \quad = \quad SlopeF_i/s - SlopeN_i, \tag{2}$$

where *SlopeN$_i$* and *SlopeF$_i$* are slopes estimated by fitting linear regression lines through plots of the $i^{th}$ intervals of $\ln[N(t)]$ and $-\ln[F(t)]$ versus *t*, respectively [17]. Note that because in the one population model the fraction of plasmid-bearing cells declines with time (Eq (6)), *slopeF* is positive and denotes the decline rate *sr* of *f(t)*. If *s* is known, by using slopes *slopeN$_i$* and *slopeF$_i$* it is straightforward to estimate the rates of Mtb replication and death from Eqs (1) and (2) as was originally proposed in Gill *et al.* [17]. The rates of Mtb replication (*r*) and death (*δ*) can be also estimated using *SlopeN$_i$* and *SlopeP$_i$* (the rate of change of log of the number of plasmid-bearing bacteria) [22].

**Estimating segregation coefficient *s* in Mtb strain HN878.** To estimate the probability of plasmid loss per division *s* we performed in vitro experiments with HN878 carrying the replication clock plasmid pBP10. In vitro we expect that bacteria do not die (i.e., $\delta = 0$) which then allows to estimate the segregation coefficient *s* using Eq (2). To do this we first calculated the total bacterial density in the cultures over time by accounting for 1:10 dilution at each transfer. Then for every growth condition/medium we calculated the growth rate of the populations *r* (by performing linear regression of $\log(N)$ vs. time and estimating *slopeN*). Then we calculated change in the percent of plasmid-bearing cells in the population over time, and calculated the decline in the percent as $\log(F)$ vs. time, i.e., we estimated *SlopeF*. Using Eq (2) with $\delta = 0$ we then estimated the plasmid loss rate as $s = slopeF/slopeN$ (for each of two media conditions separately and by pooling all the data together).

**One population model.** In order to quantify the rate of Mtb replication in mice *in vivo*, Gill *et al.* [17] proposed a mathematical model that described the change in the number of plasmid-carrying and plasmid-free bacteria with respect to time. One important assumption of the model is that all bacteria in the lung divide and die at the same rates, representing, thus, one homogeneous population (see Fig 1A). The rates of Mtb replication and death could vary with time since infection. Note that in this modeling framework we do not consider explicitly many factors that impact these rates such as availability of resources (nutrients, cells Mtb can infected, etc.) and immune response (both innate and adaptive immunity). Ability to rigorously estimate the rates of Mtb replication and death in vivo can allow then to investigate how such host factors, e.g., Mtb-specific T cell response, may influence these rates, paving the way to predict impact of vaccination on within-host Mtb dynamics.

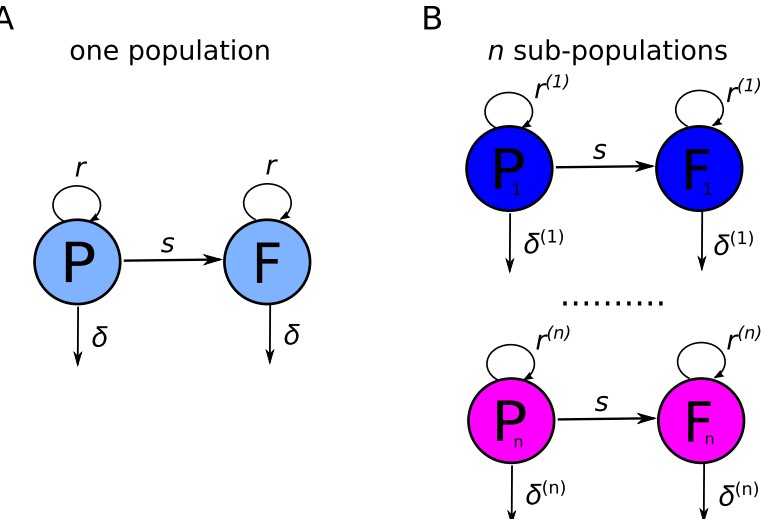

**Fig 1. Schematic of alternative mathematical models of Mtb dynamics in rabbits.** We have previously shown that dynamics of H37Rv-pBP10 bacteria in B6 mice are consistent with "one population" model in which all bacteria replicate and die at identical rates (panel **A**, [22]). In this model, plasmid-bearing (*P*) and plasmid-free (*F*) bacteria replicate and die at rates *r* and *δ*, respectively, and plasmid-free bacteria are formed during a division of plasmid-carrying cell with the probability *s*. In rabbits, due to formation of individual granulomas dynamics of bacteria is more consistent with "*n*-subpopulations" model (panel **B**). In this model, bacteria replicate and die at rates $r^{(j)}$ and $\delta^{(j)}$, respectively, where *j* represents sub-population (*j* = 1 . . . *n*); rates of replication and death may vary between sub-populations and/or with time since infection *t* (Eq (7)).

Plasmid-free and plasmid-bearing strains were assumed to have the same growth and death rates; dynamics of the cell populations are then given by the system of equations:

$$\frac{\mathrm{d}P(t)}{\mathrm{d}t} = [r(t)(1-s) - \delta(t)]P(t), \tag{3}$$

$$\frac{\mathrm{d}F(t)}{\mathrm{d}t} = [r(t) - \delta(t)]F(t) + sr(t)P(t), \tag{4}$$

$$\frac{\mathrm{d}N(t)}{\mathrm{d}t} = [r(t) - \delta(t)]N(t), \tag{5}$$

$$\frac{\mathrm{d}f(t)}{\mathrm{d}t} = -sr(t)f(t), \tag{6}$$

where $N(t) = P(t) + F(t)$ is the total number of bacteria in the lung, $P(t)$ is the number of plasmid-carrying bacteria, $F(t)$ is the number of plasmid-free bacteria, $f(t) = P(t)/N(t)$ is the fraction of plasmid-bearing bacteria in the population, $r(t)$ and $\delta(t)$ are the rates of Mtb replication and death, respectively, *t* is time in days, and the segregation probability *s* denotes the probability that a plasmid-free cell will be produced from a division of a plasmid-bearing cell. To solve the model one also needs to define the initial number of plasmid-bearing and plasmid-free bacteria, $P(0)$ and $F(0)$, respectively.

In general, the rates of Mtb replication and death may vary over time since infection, but in the absence of additional information, the simplest assumption is that the rates are constant between individual times where Mtb counts were measured but may vary between different time intervals [17, 22]. In our data, bacterial counts were measured in the lungs of rabbits at 0,

28, 56, 84, and 112 days since infection (S2 Fig), therefore, for one population model the rates of replication and death are defined as follows:

$$(r(t), \delta(t)) = \begin{cases} (r_1, \delta_1), & \text{if } 0 \leq t < 28 \text{ days,} \\ (r_2, \delta_2), & \text{if } 28 \leq t < 56 \text{ days,} \\ (r_3, \delta_3), & \text{if } 56 \text{ days} \leq t < 84 \text{ days,} \\ (r_4, \delta_4), & \text{if } 84 \text{ days} \leq t < 112 \text{ days,} \end{cases} \tag{7}$$

Model parameters $r_i$ and $\delta_i$ can be estimated by fitting the mathematical model (Eqs (3) and (4)) to data assuming a fixed value for the segregation coefficient $s$ [22]. In the previous work the probability of plasmid segregation in Mtb strain H37Rv was determined *in vitro* as $s = 0.18$ [17]. Our in vitro experiments suggested that the pBP10 plasmid is more stable in HN878 with the segregation constant being $s = 0.10$. However, we have confirmed that most of our conclusions remain valid for a higher $s = 0.18$ even though estimates of model parameters do depend on $s$ (see Results section).

**Multiple sub-populations model.** As we show in the Results section the "one population model" is not able to accurately describe dynamics of the plasmid loss from the bacterial population in rabbits over time. We propose that a non-monotonous change in the percent of plasmid-bearing cells in the bacterial population may arise due to asynchronous dynamics of several independent sub-populations of bacteria (Fig 1B). The basic mathematical model given in Eqs (3) and (4) can be easily extended by assuming $n$ independent sub-populations each starting from initial number of plasmid-bearing ($P^{(j)}(0)$) and plasmid-free ($F^{(j)}(0)$) bacteria, and each replicating and dying at rates $r^{(j)}$ and $\delta^{(j)}$, respectively, with index $j$ denoting the sub-population ($j = 1\ldots n$, Fig 1B and Eq (7)).

**Fitting ODE models to data.** One method to estimate the rates of Mtb replication and death is to use the method of linear regressions based on Eqs (1) and (2) [17]. An alternative method is to numerically solve the mathematical model given in Eqs (3) and (4) with replication and death rates defined as in Eq (7) and fit the solution to data [22]. For example, in our previous work we fitted the solution of the one population model to data on total number of bacteria and the number of plasmid-bearing bacteria in lungs of B6 mice [22]. We found, however, that while the model can reasonably well describe the dynamics of total number of bacteria and the number of plasmid-bearing bacteria, the model does not fit well the fraction of plasmid-bearing cells (e.g., see Fig 4C in McDaniel *et al.* [22]). Therefore, here we propose to use a maximum likelihood method to fit the model to the data on the total number of bacteria $N$ and the fraction of plasmid-bearing cells in the population $f$ [35, 36].

Because measurement errors in calculating a total number of bacterial in the lung and a percent of plasmid-bearing cells in the population may be different, we assume that the distribution of errors for two datasets are normal but have different variances, $\sigma_N^2$ (total cell numbers data) and $\sigma_f^2$ (fraction of plasmid-bearing cells data). For our model and the data, the general likelihood of observing the data given the model prediction on the number of bacteria found in the lung at time $t_i$, $N(t_i)$, and the fraction of plasmid-bearing cells, $f(t_i)$, is

$$L(N, f|\vec{p}) = \prod_{i=1}^{n}\prod_{j=1}^{k_i} \frac{1}{\sqrt{2\pi\sigma_N^2}} e^{\frac{-(\log \hat{N}_{ij} - \log N(t_i))^2}{2\sigma_N^2}} \times \prod_{i=1}^{n}\prod_{j=1}^{k_i} \frac{1}{\sqrt{2\pi\sigma_f^2}} e^{\frac{-(\log \hat{f}_{ij} - \log f(t_i))^2}{2\sigma_f^2}}, \tag{8}$$

where $\vec{p}$ is the vector of all model parameters, $n$ is the number of time points at which measurements were taken ($n = 5$ for our data), $k_i$ is the number of animals with measured number

of bacteria at $i^{\text{th}}$ time point (generally, $k_i = 4$), $\hat{N}_{ij}$ and $\hat{f}_{ij}$ are experimental measurements at time point $i$ and animal $j$. It is convenient to rewrite Eq (8) in terms of negative log-likelihood $\mathcal{L} = -\log(L)$:

$$\mathcal{L}(N,f|\vec{p}) = \sum_{i=1}^{n}\sum_{j=1}^{k_i}\frac{(\hat{N}_{ij} - N(t_i))^2}{2\sigma_N^2} + \sum_{i=1}^{n}\sum_{j=1}^{k_i}\frac{(\hat{f}_{ij} - f(t_i))^2}{2\sigma_f^2} + n\Big[\log(\sigma_N) + \log(\sigma_f)\Big], \quad (9)$$

where we omitted constant terms. Model parameters were estimated by minimizing negative log-likelihood $\mathcal{L}$ (Eq (9)) using `FindMinimum` routine in Mathematica 12.

**Statistics.** We compared nested models using F-test for nested models, also known as likelihood ratio test [37], and all models using Akaike Information Criterion (**AIC**) [38]. To evaluate change in the percent of plasmid-bearing cells with time we used linear regression.

**Stochastic simulations.** We simulated formation of lesions/granulomas in rabbits assuming that infection starts with a single plasmid-bearing bacterium, dividing and dying at time-invariant rates $r$ and $\delta$, respectively, in accord with Eqs (3) and (4). We used `GillespieSSA2` library in R with the following rates to simulate the dynamics stochastically using a Gillespie algorithm:

```
reactions <- list(
  #propensity function        effects            name for reaction
  reaction(~ r * PP,          c(PP = +1),          "P_division"),
  reaction(~ delta * PP,      c(PP = -1),          "P_death"),
  reaction(~ r*s * PP,        c(PP = -1,FF=+1),    "P_plasmid_loss"),
  reaction(~ r * FF,          c(FF = +1),          "F_division"),
  reaction(~ delta * FF,      c(FF = -1),          "F_death")
)
```

We used tau-leaping algorithm with $\tau = 0.1$ [39] and typically performed 1,000 simulations per parameter set. In a separate set of stochastic simulations we assumed that individual Mtb bacteria start in a resting state $R$ and become activated at a rate $\alpha$; after activation the dynamics of Mtb is governed by the same reactions as in the main model. In simulations of Mtb dynamics for 112 days, we also allowed replication and death rates to change with time since infection as estimated using one population model and implemented in GillespieSSA2 using C++ syntax.

## Results

### Estimating segregation probability of pBP10 plasmid in Mtb strain HN878 in vitro

To estimate stability of the pBP10 plasmid in Mtb strain H37Rv, Gill *et al*. [17] grew the H37Rv-pBP10 strain in vitro in different media and calculated the decline rate in the percent of plasmid-bearing cells (*slopeF*) and exponential increase rate in total cell numbers (*slopeN*) in culture over time (see Eqs (1) and (2)). Because it is expected that the bacteria do not die in culture in vitro within the experimental time frame [17], with $\delta = 0$, the segregation coefficient can be then calculated, for example, using Eq (2) as $s = slopeF/slopeN$. Clinical Mtb isolate HN878 was transformed with the replication clock plasmid pBP10 and was provided for our experiments by D. Sherman's group (University of Washington).

We grew Mtb strain HN878-pBP10 in complete 7H9 or 1:3 (7H9:PBS) diluted media in serial passage experiments (Fig 2A and see Materials and methods for more detail). In the experiment at each transfer we determined the total number of bacteria and the number of

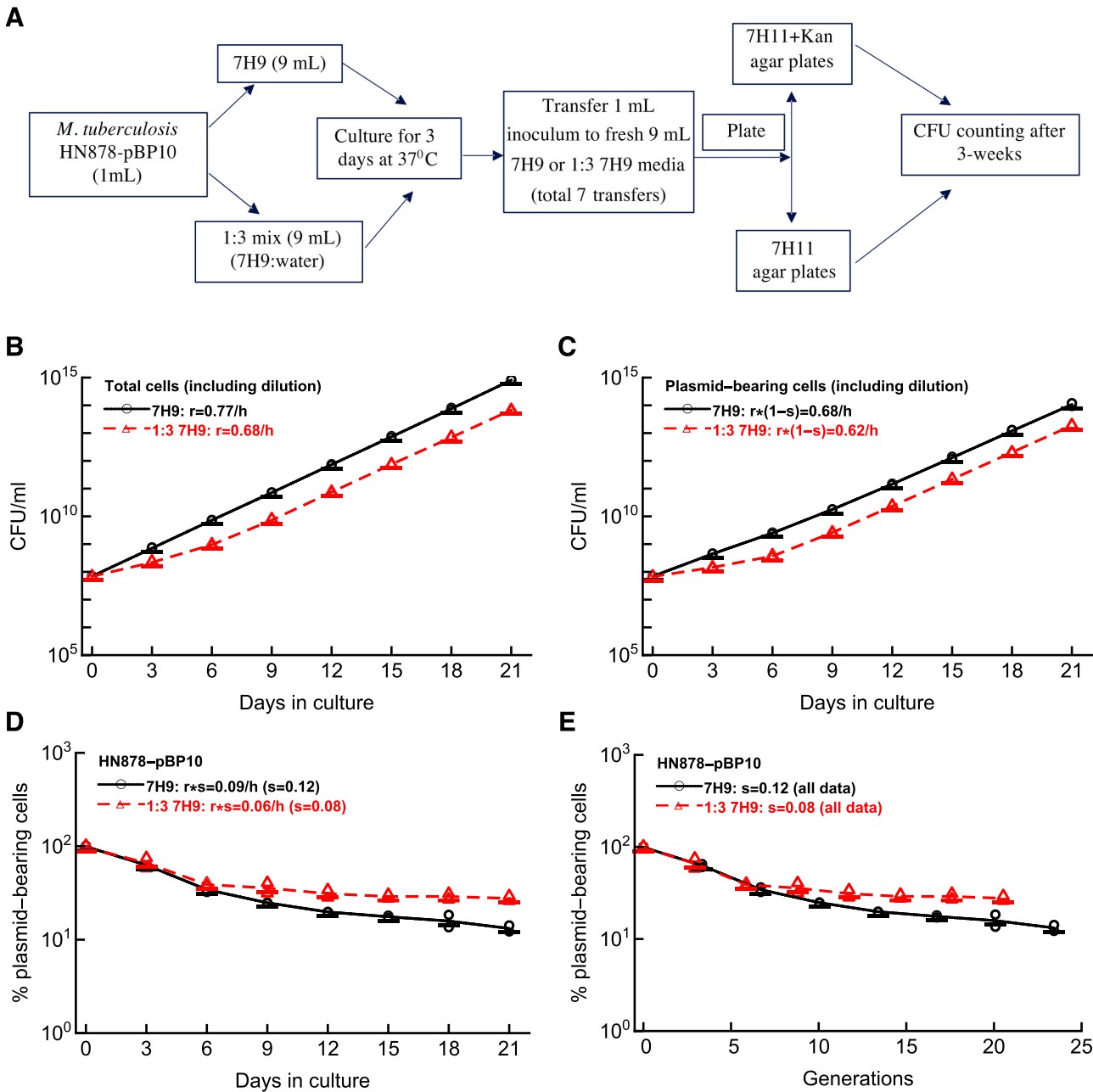

**Fig 2. Estimating the segregation coefficient *s* of the plasmid pBP10 in Mtb HN878 *in vitro*.** Clinical Mtb isolate HN878, transformed with the replication clock plasmid pBP10, was grown in vitro, and loss of the plasmid was followed over time (panel **A**). In the experiment, 1 mL of fresh culture of HN878-pBP10 was transferred into 9 mL of fresh medium every 3 days for 21 day. Two sets of media was used: complete 7H9 media and 7H9 media diluted 1:3 with PBS (see Materials and methods for more detail). We calculated the total expansion of the total bacterial population (panel **B**), expansion of the plasmid-bearing cell population (panel **C**), the change in the percent of plasmid-bearing cells in the population over time (panel **D**), or the change in the percent of plasmid-bearing cells per generation (panel **E**). In the panels, markers show individual measurements and lines connect averages per time point to guide the eye. Using linear regressions for log-transformed numbers we estimated the average rate of Mtb replication *r* (panel **B**), the rate of increase in the number of plasmid-bearing cells ($r(1-s)$, panel **C**), or the rate of decline in the percent of the plasmid-bearing cells (*sr*, panel **D**). Using Eq (1) we then calculated the segregation constant *s* (panel **D**). In panel **E** we calculated the number of generations (cell divisions) *g* in the culture for time *t* using estimated rates of Mtb replication *r* in panel **B** using relationship $g = rt/\ln(2)$.

plasmid-bearing bacteria by plating the samples on 7H11 or 7H11+kanamycin plates and calculated the bacterial numbers over time by taking into consideration 1:10 dilutions during the serial passage experiments.

As expected, the number of all bacteria and of plasmid-bearing cells increased over time reflecting the impact of serial transfers (Fig 2B and 2C). Indeed, bacterial concentration at day 3 after 10 fold dilution remained stable in complete media (S1 Fig); however, we observed a stable decline in the concentration of bacteria grown in diluted media during several initial transfers (S1 Fig). We also observed non-exponential growth of Mtb in the diluted media where initially concentration of all cells or of plasmid-bearing cells increased slowly but then after two passages, the rate of growth matched that observed for cells in complete media (Fig 2B and 2C). This was because at the end of day 3 culture before the transfer, the concentration of bacteria declined for two transfers (6 days) but then stabilized at slightly lower than $10^7$ cell/ml while the concentration of plasmid-bearing bacteria continued to decline albeit at a slower rate (S1 Fig). We do not know why the dynamics of bacterial growth changed after 6 days of culture; it may be related to the adaptation of the natural Mtb isolate HN878 to the growth in culture. This dynamics is not consistent with the continuous monotonic increase in the concentration of H37Rv-pBP10 in 7H9 media of different dilutions (e.g., Figure 1d in Gill *et al.* [17]).

As expected for an unstable plasmid pBP10, the percent of plasmid-bearing cells declined with time (evaluated in days or when re-calculated as the number of generations or cell divisions, Fig 2D and 2E) and at day 3 before each transfer (S1 Fig). We calculated generation time for the bacteria in two media (full strength 7H9 or diluted 7H9) using the relationship $g = rt/\ln(2)$ and data in Fig 2A. Interestingly, the decline in the plasmid-bearing cells per generation did not collapse into one curve as was observed previous for H37Rv-pBP10 (Figure 1f in Gill *et al.* [17]). This suggests that the kinetics of pPB10 loss from HN878 strain may be growth rate-dependent.

By using Eqs (1) and (2)) we estimated segregation coefficient to be $s = 0.12$ and $s = 0.08$ for complete and diluted 7H9 media, respectively, with the average $\bar{s} = 0.10$. When pooling all data together for the decline in the percent of plasmid-bearing cells with the number of generations the segregation coefficient per generation with 95% confidence intervals was $s = 0.106$ $(0.086 - 0.125)$. This is a smaller value than that estimated for pBP10 plasmid in H37Rv strain ($s = 0.18$, [17]) suggesting that the rate at which this unstable plasmid is lost depends on the host strain. In following analyses we used the average estimate $s = 0.10$ for HN878-pBP10.

## Non-monotonic loss of plasmid pPB10 from Mtb strain HN878 in rabbits

To study the dynamics of HN878-pPB10 in vivo, we infected rabbits with aerosolized bacteria ($\sim 500$ CFU) and measured the number of all bacteria or of plasmid-bearing cells in the lungs at different time points after the infection (Fig 3A). Importantly, we found similar numbers of bacteria in the left and right lungs even though there was a tendency of observing higher CFU numbers in the right lung (S2 Fig); this is similar to our recent observation in B6 mice [25]. In parallel, we also infected rabbits with original strain HN878 (not carrying the plasmid); we found similar CFU numbers in the lung with two Mtb strains suggesting that the plasmid pBP10 did not impact replication ability of the NH878 strain (S3 Fig).

Mtb dynamics in the lungs of rabbits was not monotonic (Fig 3B); there was a large increase in the total number of bacteria from about 500 to nearly $10^8$ in the first 28 days, then the number of bacteria declined for the next 56 days (until 84 days), and then the number of bacteria exploded to about $10^{10}$ CFU/lungs in one rabbit or was maintained at $10^7$ CFU/lungs in another rabbit (Fig 3B). Interestingly, changes in the total number of plasmid-bearing bacteria

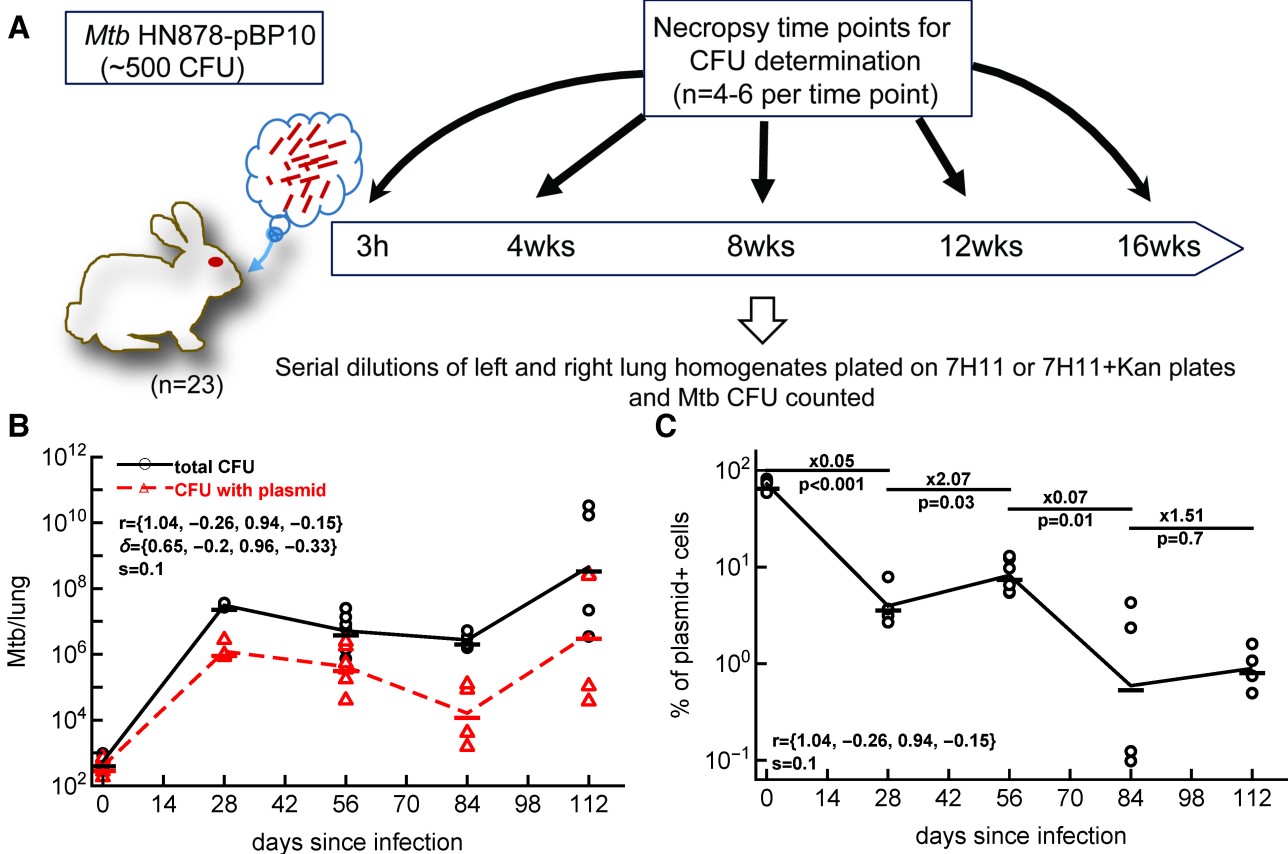

**Fig 3. Linear regression analyses suggest negative replication and death rates of Mtb in rabbits.** We infected rabbits (*Oryctolagus cuniculus*) with approximately 500 CFU of HN878 ($n = 12$) or HN878-pBP10 ($n = 11$) by aerosol. Four to six rabbits were sacrificed at 0, 28, 56, 84, and 112 days (or 0, 4, 8, 12, and 16 weeks) post-infection, and the total number of bacteria and the number of bacteria with the plasmid were determined in the right or left lung for HN878-pBP10-infected rabbits (panel **A**). Data are show for two lungs separately. We used a previously proposed method of linear regressions (see Materials and methods) and estimated the rates of Mtb replication ($r$) and death ($\delta$) in rabbits using the change in the total numbers of bacteria per lung (panel **B**) and percent of plasmid-bearing cells in the population (panel **C**). Data are shown by markers and lines connect the log-averaged data. Rates were estimated using Eqs (1) and (2) with $s = 0.10$. These estimates are for periods (0, 28), (28, 56), (56, 84), and (84, 112) days post-infection and are in per day units (listed in panel B). Because there is an increase in the percent of plasmid-bearing cells between 28 and 56 and 84 and 112 days post infection, we found negative estimates for the rates $r$ and $\delta$. Change in the percent of plasmid-bearing cells in the population was evaluated using linear regression with fold change and $p$ values from the test being indicated in panel C.

followed that of total bacteria with initial increase, decline, and again increase in the last 28 days of infection. Change in percent of plasmid-bearing cells was also non-monotonic with decline between days 0 and 28 and days 56 and 84 but with an increase between 28 and 56 days and stable levels between days 84 and 112 (Fig 3C); the latter stable frequency of plasmid-bearing cells was contrasted with the increase in the total number of bacteria in the last 28 days of the experiment (Fig 3B). This non-monotonic dynamics of the percent of plasmid-bearing cells was unexpected because we observed a monotonic decline in the percent of plasmid-bearing cells in vitro (Fig 2D) and previous studies with H37Rv-pBP10 in B6 mice also documented a monotonic loss of the plasmid with time [17, 22]. By using regression methods of Gill *et al.* [17] (Eqs (1) and (2)) with estimated $s = 0.10$, we calculated the rates of HN878 replication and death in the 4 time periods (0–28, 38–56, 56–84, and 84–112 days, Fig 3B and 3C). The analysis predicted rapid Mtb replication in the first 4 weeks of infection with $r_1 = 1.04$/day with also a relatively high death rate $\delta_1 = 0.65$/day but predicted biologically unrealistic

negative rates in second and fourth time periods. This was driven by a high increase in the percent of plasmid-bearing cells (or their somewhat stable level) in these time intervals.

## Mathematical models including two of more subpopulations of Mtb with different replication kinetics are required to explain experimental data in rabbit lungs

One of the limitations of the linear regression method of Gill *et al.* [17] is that to estimate the rate of Mtb replication and death one must use a pair of sequential data points. In the case when the percent of plasmid-bearing cells increases with time, the method provides negative estimates for the Mtb replication and/or death rate which is biologically unrealistic. We have previously developed an approach to fit the model of Mtb dynamics to the data for the whole experiment but assuming that the model parameters depend on time period between individual times when CFU were measured [22, see Eqs (3)–(6) and (7)]. In this approach we can restrict the parameters of the model to be non-negative, and we can then compare the quality of the model fit as compared to alternative models. There are different ways of how this model (Eqs (3) and (4)) can be fitted to data, and we have previously fitted the model predictions to the number of total bacteria and plasmid-bearing bacteria in murine lungs [22]. An alternative approach is to fit the model to the data on the total number of bacteria and the percent of plasmid-bearing cells in the population using a likelihood approach (Eqs (8) and (9)). In our following analyses we chose this latter, likelihood-based approach as it allows to stronger penalize models that do not accurately describe changes in the percent of plasmid-bearing cells.

We then fitted the basic model of Mtb dynamics with time-dependent parameters to the data while constraining the parameters to be non-negative. Interestingly, the model could relatively well describe the dynamics of the total number of bacteria and of plasmid-bearing bacteria (Fig 4A); however, the model did not accurately match the change in the percent of plasmid bearing cells in two time periods (Fig 4C) and predicted no Mtb replication between 28 and 56 days post infection. Taken together, two alternative approaches that we and others have used to accurately describe the dynamics of pBP10-containing H37Rv strain of Mtb in B6 mice could not explain the data on HN878-pBP10 growth dynamics in rabbit lungs.

There may be several potential reasons for an increase in the percent of plasmid-bearing cells between 28 and 56 days post-infection. One possibility is an experimental error in measuring the percent of plasmid-bearing cells. However, because with the exception of data for 84 days post-infection, measurements of percent of plasmid-bearing cells have relatively low variance, we believe this explanation is unlikely. Another possibility is that plasmid-free cells may acquire the plasmid from the environment by transformation. However, given difficulty at transforming plasmids into Mtb strains, we also do not believe this is a viable explanation. Finally, it is possible that our mathematical modeling approach makes assumptions that break for Mtb growth dynamics in rabbit lungs; specifically, the assumption that all bacterial cells replicate and die at the same rates within the host. Indeed, it has been previously observed that Mtb infection of rabbits results in formation of heterogeneous granulomas, special structures of bacteria and various host cells [26, 29, 32, 40–42]. It is also possible that Mtb dynamics may not be fully synchronized in different granulomas, and indeed, PET/CT imaging has suggested that in monkeys, different Mtb granulomas can have asynchronous growth dynamics [30].

Therefore, we extended our initial model to include two or three bacterial sub-populations with different replication and death rates (Fig 1) and fitted different versions of such a model to the data on HN878-pBP10 dynamics in rabbits. Modeling each additional sub-population requires at most 10 parameters (2 parameters for the initial condition and 8 parameters for replication and death rates for four time intervals) and it may be easy to overfit the model to

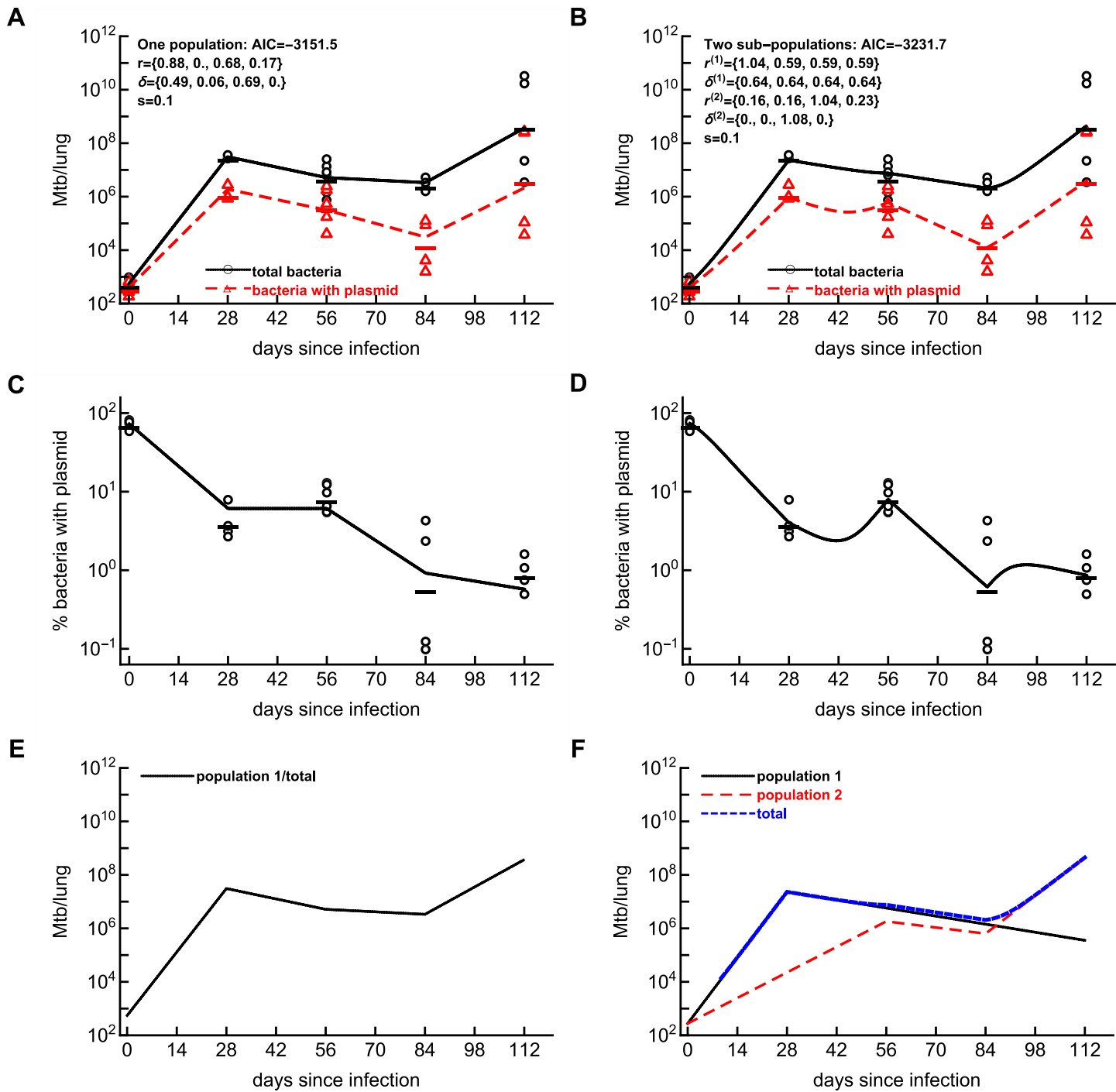

**Fig 4. Mathematical model with at least two sub-populations is required to explains Mtb dynamics in rabbits.** We fit the one population (panels **A**, **C**, **E**) or two sub-populations (panels **B**, **D**, **F**) dynamics model (based on Eqs (3) and (4)) to the data on Mtb HN878-pBP10 dynamics in rabbit lungs using a likelihood approach (see Materials and methods and Eq (9)) and estimate the rates of Mtb replication and death assuming that all parameters are non-negative. Because of many different ways of how dynamics of bacteria in two different sub-population may occur, we constrained as many parameters as possible to not vary with time since infection. We show the data on the dynamics and model fits of the total number of bacteria in the lungs (A-B), the percent of plasmid-bearing bacteria (C-D). The data are shown by symbols and model fits by lines. Panels E-F show model predictions on the dynamics of the total population (E) or sub-populations (F) in the models. Note that panels A-B also include the data and model prediction on the number of plasmid-bearing bacteria in the lungs (triangles/dashed lines) which were not used in model fitting. Estimated model parameters are shown in panels A-B; other parameters are for one population model: $P(0) = 394.6$, $F(0) = 150.6$, $\sigma_N = 0.172$, $\sigma_f = 0.081$, and the negative log-likelihood of the fit was $\mathcal{L} = -1587.74$. For two sub-population model: additional parameters are $P^{(1)}(0) = P^{(2)}(0) = 204.2$, $F^{(1)}(0) = F^{(2)}(0) = 72.6$, $\sigma_N = 0.17$, $\sigma_f = 0.073$, $\mathcal{L} = -1628.84$. In total, one and two population models have 12 and 13 fitted parameters, respectively.

data. We therefore assumed the model with two sub-populations while constraining the parameters for the two sub-populations to be same as much as possible. For example, we assumed identical initial number of plasmid-bearing and plasmid-free bacteria in different sub-populations. Interestingly, such a model could well describe experimental data on the dynamics of all bacteria and plasmid-bearing bacteria, and it could reasonably well predict increase in the percent of plasmid-bearing cells between 28 and 56, and 84 and 112 days post infection in rabbit lungs (Fig 4B and 4D) with only 13 parameters (including 2 parameters for the errors $\sigma_N$ and $\sigma_f$). According to the model fit, Mtb population consists of at least two sub-populations of bacteria. One sub-population rapidly replicates early during the infection ($r_1^{(1)} = 1.04$/day and $\delta_1^{(1)} = 0.64$/day) and thus few bacteria in this sub-population have the plasmid. This sub-population is then controlled by immunity and contracts while still dividing ($r_2^{(1)} = 0.59$/day and $\delta_2^{(1)} = 0.64$/day) and further losing the plasmid over time (Fig 4F). Another sub-population replicates more slowly early in infection ($r_1^{(2)} = 0.16$/day and $\delta_1^{(2)} = 0$) and many bacteria still have the plasmid by 56 days post-infection (Fig 4F). Between 56 and 84 days, this second sub-population replicates rapidly ($r_3^{(2)} = 1.03$/day and $\delta_3^{(2)} = 1.07$/day) resulting in the loss of the plasmid in the population but because total numbers of bacteria still decline rapid replication is balanced by a relatively high death rate. Finally in the last time interval, continued growth of the second sub-population ($r_4^{(2)} = 0.23$/day and $\delta_4^{(2)} = 0$) results in increase in the number of bacteria in the lung while slightly increasing the percent of plasmid-bearing cells in the population. Therefore, the fit of the model with two sub-populations is significantly better than that with one kinetically homogeneous population ($\chi_1^2 = 82.2$, $p \ll 0.001$ and $\Delta AIC \gg 10$). Visually, the two-subpopulation model now accurately describes non-monotonic change in percent of plasmid-bearing cells, something that one population model failed to do.

There are clear reasons of why at least two sub-populations with different replication kinetics are required to accurately describe the change in the number of bacteria and the percent of plasmid-bearing cells. First, the model must explain large decline in the percent of plasmid-bearing cells and large increase in the total number of bacteria in the lung during first 28 days (Fig 4B&4D). One population of bacteria can explain such data (Fig 4A&4C). Then to explain the increase in the percent of plasmid-bearing cells, while the total number of bacteria declines between 28 and 56 days post infection, one must assume that the initial dominant population contracts which another population with a higher frequency of plasmid-bearing cells (due to fewer divisions) becomes more dominant. Large decline in the percent of plasmid-bearing cells and yet also a decline in the total number of cells between 56 and 84 days post-infection implies rapid cell division that must be counterbalanced by high death rate (Fig 4B, 4D and 4F). Finally, a large increase in the number of bacteria with relatively constant percent of plasmid-bearing cells between 84 and 112 days post-infection suggest expansion of previously poorly replicating/subdominant sub-populations with a high initial frequency of plasmid-bearing cells.

The model with two sub-populations can describe the Mtb dynamics data in rabbit lungs relatively well. However, predictions of the model of the dynamics of plasmid-bearing cells seem puzzling with a sharp peak at day 56 and a sharp nadir at day 84 suggesting that the model may not be fully biologically realistic (Fig 4D). We therefore investigated if a model with three sub-populations may describe the data more reasonably. This was a challenging exercise since we started with a model with 30+ parameters that clearly overfitted the data. By constraining many of the rates to be same between different time periods (e.g., as we have done previously [22]) as well as by assuming the same initial numbers of cells in different

subpopulations we could find sets of parameters that fitted the data better than all our previous models but with a relatively small number of parameters (e.g., in S4 Fig we used 13 parameters). In this model, the third population that was declining until day 84 was needed to explain a large increase in the total number of plasmid-bearing bacteria while maintaining a relatively high frequency of plasmid-bearing cells by 112 days post infection (S4 Fig). Interestingly, while the three population model fitted the data statistically better than either a single population model ($\chi_1^2 = 134.0$, $p \ll 0.001$) or two population model ($\Delta$AIC > 10), it still predicted non-monotonic dynamics of plasmid-bearing cells after 56 days post-infection (S4 Fig). This result suggests that a more continuous model prediction of the dynamics of percent of plasmid-bearing cells may require having more sub-populations and/or more parameters.

## Alternative analyses to check robustness of our conclusions

When fitting models with two or three sub-populations, we typically started with many parameters and fitting such a model to data often resulted in overfitting unless we reduced the number of fitted parameters. Obviously, there are many different ways to reduce the number of model parameters and we could come up with several different parameter sets that could reasonably well describe the data. This exercise suggested that the specific parameters inferred by fitting the model with several independent Mtb sub-populations to data (e.g., those in Fig 4B) need to be interpreted with caution.

Infection of monkeys and mice with very low doses of barcoded Mtb suggested that individual lung lesions are started by single bacteria [25, 30, 43]. Due to a low number of starting bacteria, Mtb dynamics in individual lesions is likely to be stochastic. As suggested by a reviewer we therefore investigated whether stochasticity Mtb dynamics in individual lesions in accord with one population model may explain non-monotonic change in the percent of plasmid-bearing cells in the whole lung. We therefore ran Gillespie simulations of Mtb dynamics for 112 days for 545 bacteria (394 and 151 plasmid-bearing and plasmid-free cells, respectively, as was observed on average in our experimental data, Fig 3B) and calculated the percent plasmid-bearing cells and total number of cells in the lung (S5 Fig). Even though Mtb dynamics in each lesion was highly stochastic (see next section), difference in total number of bacteria from 545 starting lesions was small and could not explain the data accurately (S5 Fig). We extended the model to allow for bacteria to start infection with a "resting" state *R* and become activated at a rate $\alpha$. Interestingly, at very small activation rate (e.g., $\alpha = 0.005$/day) change in the percent of plasmid-bearing cells became highly stochastic even for 545 initial bacteria, in part, because only a small proportion of the initially deposited bacteria became activated (S5 Fig). However, model simulations still were not able to explain increase in the percent of plasmid-bearing cells between 28 and 56 days and 84 and 112 day post-infection (S5 Fig). Furthermore, because in simulations only few cells became activated, the model could not accurately describe the dynamics of total number of bacteria in the lung (S5 Fig). Matching dynamics of the total number of bacteria in the lung observed experimentally would require increasing the rate of Mtb replication beyond what is considered to be realistic (about 1–1.2/day). We therefore conclude that stochasticity of Mtb dynamics in one population model is not able to explain the non-monotonic change in the percent of plasmid-bearing cells in rabbit lungs.

To fit models to data we log-transformed the data and model predictions for the total cell number and the percent of plasmid-bearing cells while assuming that these two sets of measurements may have different measurement error. While using log transformation for the percent of plasmid-bearing cells is similar to logit transformation at small percents and thus is appropriate, we repeated some of our analyses using arcsin(sqrt) transformation for frequencies that also may normalize residuals for proportions [44, 45]. We found that as with log

transformation one sub-population model was not able to accurately describe the dynamics of percent of plasmid-bearing cells in the population, while two sub-population model significantly improved the fit both statistically ($\chi^2_1 = 98.97$, $p < 0.001$) and visually.

We noted that at the last experimental time point (112 days post infection) Mtb growth dynamics in rabbit lungs follow a divergent pattern—in one animal total CFU numbers remained stable at about $10^8$ bacteria, and in another animal they exploded to about $10^{10}$ bacteria per lung (Fig 4A and 4B). We reasoned that describing such dynamics using average parameters may not be fully appropriate. Therefore, we repeated our analyses for the data including only measurements up to 84 days. Importantly, we found that a model with a single/ homogeneous Mtb population cannot explain Mtb dynamics accurately and including at least 2 sub-populations with different replication/death rates is required to adequately describe the data (S6 Fig).

In our experiments we found that the plasmid segregation coefficient ($s = 0.10$) is lower than that measured in H37Rv in a previous study ($s = 0.18$) [17]. Previously we have shown that value of the segregation coefficient is critical at determining the actual values of Mtb replication and death rates [22]. We therefore investigated if a higher segregation coefficient is consistent with our experimental data on Mtb growth dynamics in rabbits. Interestingly, the model with three sub-populations and $s = 0.18$ was able to explain the Mtb growth dynamics with the same quality as the model with $s = 0.10$ (S7 Fig). Importantly, however, increase in the segregation coefficient resulted in lower estimated Mtb replication and death rates (e.g., $r = 0.58$/day and $\delta = 0.15$/day for the first sub-population in the first 28 days post-infection) as expected [22]. Therefore, it is critical to make sure that estimates of the segregation coefficient $s$ are as precise as possible.

## Identifying conditions to inform replication history of Mtb in individual granulomas of rabbits

Our finding that experimental data on Mtb dynamics in rabbits can be better explained by models of heterogeneous sub-populations suggests that tracking Mtb dynamics in individual lesions may help explain why the number of bacteria recovered from different lesions varies orders of magnitude [30]. However, because it is currently believed that individual lesions are likely to be formed by single bacteria (e.g., [25, 30, 43]) only lesions established by plasmid-bearing cells would inform about Mtb replication and death rate in the lesion if we are to use an Mtb strain with pBP10 plasmid. To investigate whether typical sampling of Mtb in individual lesions in rabbits 3–4 weeks post-infection (21–28 days) is informative, we ran Gillespie simulations of Mtb dynamics starting with one plasmid-bearing cell and for different rates of Mtb replication $r$ and death $\delta$ in this time period (see Materials and methods, and Fig 5 and S8 Fig).

Consistent with our recent results [25], starting infection with a single bacterium often results in extinction, for example, with $r = 1.04$/day and $\delta = 0.64$/day, only $405/1000 \approx 40\%$ runs resulted in lesions with Mtb counts above 0, and only 30% of the runs resulted in lesions containing plasmid-bearing cells above the limit of detection (Fig 5A and 5B). Most extinctions occurred when Mtb counts were low, so they would not be detected experimentally. The dynamics of plasmid-bearing cells was also highly variable between individual runs with about 25% of runs (at $r = 1.04$/day and $\delta = 0.64$/day) resulting in the loss of the plasmid from the population within 28 days (Fig 5C). Importantly, while the total number of bacteria remained relatively constant for these parameter combinations ($r - \delta = 0.4$/day), the percent of plasmid-bearing cells was progressively lower for a higher Mtb replication rate, consistent with the prediction of the deterministic model (Fig 5D and Eq (6)). Furthermore, in simulations with

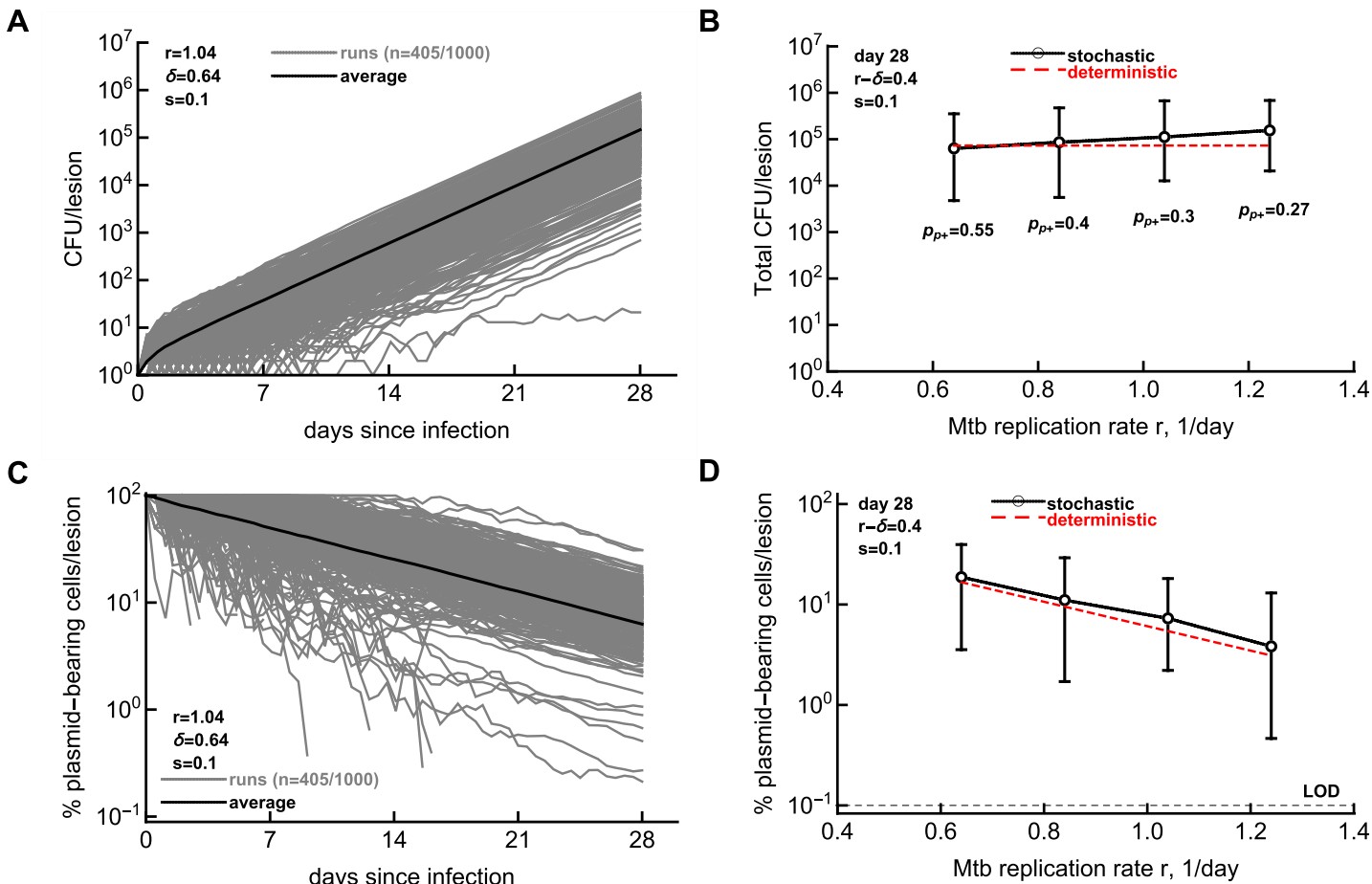

**Fig 5. Using replication clock plasmid to infer replication history of granulomas with similar bacterial numbers.** We simulated Mtb dynamics in individual lesions starting with one plasmid-bearing cell and varied the rates of Mtb replication ($r$) and death ($\delta$). We show the trajectories from 1000 simulations of the total number of bacteria per lesion (panel **A**) or the percent of plasmid-bearing cells in the lesion (panel **C**). Parameters of the simulations were $r$ = 1.04/day, $\delta$ = 0.64/day, and $s$ = 0.1 (per Fig 4B). For these parameters, 405 simulations did not result in extinction. Note that in some cases in panel C, frequency of plasmid-bearing cells drops to 0 due to loss of the plasmid in the population; these 0-values are excluded in the graph. Next, we varied the rates of Mtb replication and death so that the net rate of Mtb replication remains the same, $r - \delta$ = 0.4/day. We selected only simulations in which lesions contained plasmid-bearing cells above the limid of detection (**LOD**) of 0.1%. The probability of having a lesion with plasmid-bearing cells above LOD $p_{P+}$ is shown in B. We show the average and 95% confidence intervals of total number of bacteria per lesion (panel **B**) or the average percent of plasmid-bearing cells per lesion (panel **D**) at 28 days (4 weeks) post-infection. Dashed red lines in B&D denote predictions of the deterministic mathematical model ($N(t) = e^{(r-\delta)t}$ and $f(t) = e^{-rst}$, respectively).

increased bacterial replication rate and a constant death rate resulted in progressively lower percent of plasmid-bearing cells (S8 Fig); however, keeping bacterial replication rate constant while varying death rate predicted similar percent of plasmid-bearing cells (S8 Fig). This result suggests that measuring percent of plasmid-bearing cells in individual granulomas of rabbits should allow to estimate the rate of Mtb replication, and thus, should indicate if lesions with more bacteria also have evidence of more cycles of replication. Measuring lesion-localized T cell response may also help clarify how immunity influences Mtb replication and death rates. Additionally, by measuring the frequency of granulomas containing plasmid-bearing cells (and comparing this number with the percent of plasmid-bearing cells shortly after aerosol infection), we should be able to estimate the average rate of Mtb replication in the whole animal.

## Discussion

In this study we followed the kinetics of replication of a pathogenic Mtb strain HN878 containing an unstable, "replication clock" plasmid pBP10 in rabbit lungs. We determined that this plasmid is lost from HN878 in vitro with a segregation coefficient $s = 0.1$ (Fig 2) suggesting that the plasmid is more stable in this strain as compared to H37Rv [17]. Interestingly, the dynamics of percent of plasmid-bearing cells in rabbit lungs was not monotonic as expected from previous studies [17, 22] increasing between 28 and 56 days post-infection (Fig 3). We showed that a previously suggested model assuming a single, kinetically homogeneous Mtb population is not consistent with these data, and including two and three sub-populations with different replication/death kinetics would allow the model to more accurately fit these data (Fig 4 and S4 Fig). Simulating Mtb dynamics stochastically assuming one-population model in which individual bacteria become activated at different times after infection did not allow to accurately match experimental data (S5 Fig) suggesting that stochastic dynamics of one population model cannot explain non-monotonic change in the percent of plasmid-bearing cells in rabbit lungs. Stochastic simulations also suggested that measuring the percent of plasmid-bearing cells in individual granulomas of rabbits should allow to estimate Mtb replication and death rates in the granuloma (Fig 5 and S8 Fig), and thus, should provide quantitative explanation of why different granulomas have dramatically different numbers of viable bacteria.

As far as we are aware, our study is the first one to provide quantitative estimates of how quickly pathogenic Mtb strain HN878 replicates in rabbit lungs after aerosol infection. It is difficult to directly compare our estimates with those published previously (e.g., [17, 22]) given that we found that previous models do not adequately describe our experimental data. The highest replication rate we estimated was $r = 1.04$/day, higher than that for H37Rv in mice ($r = 0.72$/day in McDaniel *et al.* [22]). There are other methods used to evaluate how quickly Mtb replicate in vivo including CEQ or RS ratio [18, 23, 30]; however, these methods remain qualitative and do not provide rigorous estimates of Mtb replication rates, particularly in heterogeneous in vivo granulomas. Future studies may need to compare Mtb replication kinetics using these different methods to identify which methods are most accurate.

Following an analysis performed by Gill *et al.* [17] we calculated he total number of bacteria produced and died in each of two sub-populations $j = 1, 2$ in our two-subpopulation model as $\int_0^{112} r^{(j)}(t)N^{(j)}(t)\mathrm{d}t$ and $\int_0^{112} \delta^{(j)}(t)N^{(j)}(t)\mathrm{d}t$, respectively (see Eq (5) and Fig 1 in Materials and methods for detail). The total number of bacteria produced in 1st and 2nd subpopulations are $3.3 \times 10^8$ and $4.8 \times 10^8$, respectively (total is $8.1 \times 10^8$), and the total bacteria that died during the experiment are $3.3 \times 10^8$ and $0.3 \times 10^8$, respectively (total is $3.6 \times 10^8$). These calculations highlight that second subpopulation contributes most of the bacteria produces during the infection and is associated with less death, explaining how this sub-population drives disease in the animals (due to a large increase in CFU in some animals, Fig 4A&4B). Interestingly, Gill *et al.* [17] calculated that $1.3 \times 10^7$ CFU/ml of total bacteria or, assuming a lung weight of 150mg, about $2.6 \times 10^6$ are bacteria produced in the whole lung of B6 mice infected with H37Rv over 111 days of infection. We conclude here that HN878-pBP10 in rabbits undergoes a larger number of replications than a more commonly used lab strain H37Rv in mice further highlighting differences between these Mtb strains and host species.

Treating Mtb-infected rabbits with antibacterial drugs revealed differential response of different lung granulomas to the drugs suggesting heterogeneity in Mtb replication in rabbits [46]. By using the RS ratio metric Walter *et al.* [47] elegantly showed differences in Mtb replication rates in different lung areas of C3HeB/FeJ mice that form well-circumscribed necrotic/caseating granulomas [48, 49]. Here we extend these studies by showing that even when measuring Mtb numbers in the whole lung of rabbits, along with the percent of bacterial cells

carrying plasmid pPB10, we can detect and to a degree quantify heterogeneity in rates of Mtb replication and death in vivo. Our results thus may further help explain the differential response of Mtb-infected rabbits to drug treatment and may help design better treatment strategies depending on the distribution of granulomas with slow or rapidly replicating bacteria.

Our work has several limitations. Estimated segregation coefficient *s* of the plasmid pBP10 in HN878 strain differed slightly between two culturing conditions and the plasmid loss kinetics was changing over time (Fig 2). This could be due to the inherent nutritional composition of the growth media used (i.e, full-strength versus diluted 7H9 media). Additional experiments to measure the instability of this plasmid in more diverse growth conditions in vitro would be useful.

We have used a relatively large inoculum of Mtb ($\sim$ 500 CFU/rabbit) that may be in contrast with murine studies using smaller infectious doses (e.g., [25]). For Mtb infection studies such as in our lab, 6–8 weeks old mice have been commonly used. Considering the lung of mice with about 150 mg in weight and $\sim$ 1 ml total lung capacity (**TLC**) at this age, a conventional dose of 100 CFU is a typical inoculum [25]. In comparison, rabbits used in our studies were about 2.2 to 2.6 kg in weight, with a lung weight of about 15 grams and a TLC of about 62 ml. Thus, the rabbits we used have about 100x more lung weight and 60x more TLC than the mice. The equivalent of 100 CFU in mice would be either 10,000 CFU (based by weight) or 6,300 CFU (based on TLC) for the rabbit infection. However, we used only 500 CFU/rabbit, which is just 5x more in absolute numbers as compared to conventional dose used in murine infections. We believe this is a smaller inoculum than what has been used in a typical mouse Mtb infection study.

We detected an increase in the percent of plasmid-bearing cells between 28 and 56 days post-infection and interpreted it as being due to heterogeneous dynamics of Mtb in rabbits. However, other possibilities exist. For example, measurements of the percent of plasmid-bearing cells have experimental errors and such an increase may arise due to such error. Given that for each sample, the percent of plasmid-bearing cells is measured twice and that we found consistent percent of plasmid-bearing cells between different animals, we believe that this explanation is unlikely. Plasmid pBP10 may theoretically be taken up by plasmid-free cells that would then convert into plasmid-bearing cells. However, given that clinical Mtb isolates do not carry plasmids and of difficulty of transforming plasmids into Mtb, this is also an unlikely hypothesis.

The key assumption in our analysis is that the segregation coefficient *s* is independent of the rate of Mtb replication, so that the percent of plasmid-bearing cells declines at a rate proportional to the replication rate (Eq (6)). While the evidence that *s* = *const* in vitro is relatively strong (Fig 2E and [17]), it is still unknown whether *s* varies with time or between different Mtb sub-populations in vivo. We have previously shown that change in segregation coefficient *s* with time dramatically impacts estimates of the rates of Mtb replication and death [22]. Future studies will need to use different methods (e.g., based on replication clock plasmid, RS ratio or CEQ measurements, [17, 18, 23, 30]) and evaluate whether these different metrics give similar estimates of Mtb replication rates and death in vivo.

While we propose that to explain the data, Mtb population must consist of several (at least 2) semi-independent sub-populations, we do not know what these sub-populations may represent. One possibility is that these sub-populations may reside in different areas of the lung with different access to oxygen, other nutrients, or different access by the immune response [47]. It also could arise from different subpopulations within individual granulomas, e.g., cells may have different replication kinetics when they are intracellular vs. extracellular. Heterogeneity may arise due to segregation coefficient *s* being variable in different Mtb sub-populations. Of course, it is likely that the number of different sub-populations could be even larger that two

or three, e.g., if individual bacteria seeding different lung tissues have somewhat independent dynamics as has been proposed for Mtb dynamics in non-human primates [30]. However, our current data do not allow to rigorously evaluate presence of tens to hundreds of sub-populations due to risk of overfitting the models to data. Using Mtb strains with the replication clock plasmid, sorting different cell sub-populations and using other methods to evaluate Mtb replication rate (e.g., [50]) may help discriminate between these alternatives.

To have a large population of plasmid-bearing cells, we infected rabbits with a relatively high dose of Mtb that likely exceeds the dose of Mtb that humans are typically exposed to acquire stable infection [25]. How the dynamics would proceed at lower doses is uncertain although we made model-driven predictions on how parameters of Mtb replication rate and death may influence the size and composition of Mtb replicating in individual granulomas of rabbits (or other animal species) starting with a single plasmid-bearing cell. While we did use stochastic simulations, we did not derive analytical predictions of how Mtb replication/death rates would impact the distribution of the total and the percent of plasmid-bearing cells in heterogeneous granulomas. This remains an important direction of our research.

The models we have developed and fitted to data lack many biological details about Mtb dynamics in vivo. For example, it is well understood that in granulomas, Mtb residing in different regions (i.e., inside of macrophages, in caseum, etc) may have different replication capacities [47]. While we estimated the rates of Mtb replication and death we did not investigate how host's immune response controls these rates. In part, this is because measurements of Mtb-specific immune responses was not done in our experiments. To gain better insights into how immunity controls Mtb dynamics, our future work will focus on correlating estimated replication and death rates with magnitude of immune response components in individual granulomas of rabbits.

Our work opens avenue for future research of Mtb growth dynamics in TB granulomas. Transforming pBP10 plasmid into other clinical Mtb isolates would allow to compare how these strains replicate in various animal models. The percent of plasmid-bearing cells indicates cumulative replication history of bacteria, while other metrics such as RS ratio are instantaneous measures of Mtb replication [23]. Generating quantitative link between these two metrics would allow to better understand processes regulating size and complexity of individual granulomas in vivo. By measuring dynamics of plasmid loss in individual granulomas in rabbit lungs we should be able to determine if granulomas with more rapidly dividing bacteria are related to the location of these granulomas in the lung and/or due to local immune response. In humans, TB patients typically have cavities in upper and back parts of the lung and importance of lesion location in rabbit lungs for TB has been highlighted before [51–53]. Considering that the rabbit model of pulmonary Mtb infection can produce cavitary granulomas in the lungs, correlating Mtb replication kinetics in individual granulomas in control and vaccinated animals can help identify immune components that control Mtb growth, helping with development of the next generation of TB vaccines.

## Supporting information

**S1 Fig. Kinetics of HN878-pBP10 over several transfers in serial in vitro culture.** We performed new experiments in which Mtb strain HN878-pBP10, carrying an unstable plasmid pBP10, was cultured in complete 7H9 (panels **A**, **C**, **E**) or 1:3 diluted 7H9 (panels **B**, **D**, **F**) media for 3 days and transferred to new media. We determined concentration of the total number of bacteria (A-B), of the number of plasmid bearing cells per mL (C-D), or the percent of plasmid-bearing cells in the population (E-F) at the end of each 3 day culture.
(PDF)

**S2 Fig. Kinetics of Mtb replication is similar in the left and right lungs of rabbits.** We plot the data are for the total number of bacteria and number of plasmid-bearing bacteria in the lungs (panels **A** and **B**) or the percent of plasmid-bearing cells in the population (panel **C**) separately for the left and right lung. Markers denote measurements for individual rabbit lungs and lines connect geometric means of bacterial counts.
(PDF)

**S3 Fig. Mtb strain HN878-pBP10 has similar growth capacity in vivo as the original strain HN878.** We infected rabbits with HN878 or HN878-pBP10 and measured bacteria numbers in the whole lung at different days after infection. We plot average total number of bacteria in lungs for all rabbits (panel **A**) or compare lung CFU for individual rabbits (panel **B**).
(PDF)

**S4 Fig. A model with three sub-populations can more accurately explains Mtb dynamics over the course of 112 days (16 weeks) of infection.** We extended the basic, one population model to include three asynchronous sub-populations of bacteria and fitted the model to the data as described in Fig 4. Label notations are the same as in Fig 4. Estimated parameters are show in panel A, additional parameters are $P^{(1)}(0) = P^{(2)}(0) = P^{(3)}(0) = 131.5$, $F^{(1)}(0) = F^{(2)}(0) = F^{(3)}(0) = 50.3$, $\sigma_N = 0.17$, $\sigma_f = 0.073$, $\mathcal{L} = -1633.83$, AIC $= -3229.53$.
(PDF)

**S5 Fig. Stochasticity in dynamics of bacteria in individual granulomas cannot explain the non-monotonic change of the percent of plasmid-bearing cells with time.** We simulated Mtb dynamics starting with 545 bacteria out of which 394 had the plasmid (as was observed in our experimental data) using parameters estimated by fitting one population model to the data (Fig 4A). In simulations we assume that bacteria start replicating and dying at day 0 (panel **A&B**) or that infection starts with bacteria in resting state and these bacteria become activated over time at a rate $\alpha = 0.005$/day (**C&D**). We show the data (markers) and model predictions for the dynamics of 545 bacteria (gray lines for individual simulations and solid black lines are the averages) for the percent of plasmid-bearing cells (A&C) or the total number of bacteria in the lung (B&D). Simulations for 545 independent bacteria were repeated $n = 100$ times.
(PDF)

**S6 Fig. Mathematical model with at least two sub-populations is required to explains Mtb dynamics in the first 84 days since infection.** We fitted one population (panels **A, C, E**) or two sub-population (panels **B, D, F**) models to the data on Mtb dynamics in rabbit lungs for the first 84 days since infection using a likelihood approach (see Fig 4). Estimated model parameters for one population model are shown in panel A; other parameters are $P(0) = 393.4$, $F(0) = 151.8$, $\sigma_N = 0.081$, $\sigma_f = 0.091$. The negative log-likelihood of the fit was $\mathcal{L} = -1267.65$. While fits in panel A look reasonable, the model is unable to accurately describe increase in the percent of plasmid-bearing cells between 28 and 56 days since infection (panel B). Estimated model parameters for two sub-populations model are shown in panel B; other parameters are $P^{(1)}(0) = P^{(2)}(0) = 197.2$, $F^{(1)}(0) = F^{(2)}(0) = 75.5$, $\sigma_N = 0.081$, $\sigma_f = 0.087$, $\mathcal{L} = -1282.78$. The fit with two sub-populations is significantly improved as compared to the fit with one population model ($\chi_1^2 = 30.3, p \ll 0.01$).
(PDF)

**S7 Fig. A model with three sub-populations explains well Mtb dynamics over the course of 112 days (16 weeks) of infection with a higher segregation coefficient *s*.** We extended the basic, one population model to include three asynchronous sub-populations of bacteria and fitted the model to the data as described in Fig 4 assuming that $s = 0.18$. Label notations are the

same as in Fig 4. Estimated parameters are show in panel A, additional parameters are $P^{(1)}(0) = P^{(2)}(0) = P^{(3)}(0) = 131.5$, $F^{(1)}(0) = F^{(2)}(0) = F^{(3)}(0) = 50.3$, $\sigma_N = 0.17$, $\sigma_f = 0.073$, $\mathcal{L} = -1634.04$.
(PDF)

**S8 Fig. Percent of plasmid-bearing cells indicates the extend of Mtb replication in individual granulomas.** We performed simulations similarly as in Fig 5 but by varying replication rate $r$ or death rate $\delta$ independently. We selected trajectories in which the percent of plasmid-bearing cells was above LOD. We show how the total number of bacteria (panels **A**-**B**) or the percent of plasmid-bearing cells (**C-D**) per lesion changes with increasing replication rate (A&C) or death rate (B&D). Bacterial numbers are sampled at 28 days (4 weeks) post-infection.
(PDF)

## Acknowledgments

We would like to thank David Sherman for providing Mtb strain HN878-pPB10 for our experiments.

## Author Contributions

**Conceptualization:** Vitaly V. Ganusov, Selvakumar Subbian.

**Data curation:** Vitaly V. Ganusov, Selvakumar Subbian.

**Formal analysis:** Vitaly V. Ganusov.

**Funding acquisition:** Selvakumar Subbian.

**Investigation:** Afsal Kolloli.

**Methodology:** Vitaly V. Ganusov, Afsal Kolloli, Selvakumar Subbian.

**Project administration:** Vitaly V. Ganusov, Selvakumar Subbian.

**Resources:** Selvakumar Subbian.

**Software:** Vitaly V. Ganusov.

**Supervision:** Selvakumar Subbian.

**Visualization:** Vitaly V. Ganusov, Selvakumar Subbian.

**Writing – original draft:** Vitaly V. Ganusov, Afsal Kolloli.

**Writing – review & editing:** Vitaly V. Ganusov, Selvakumar Subbian.

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
