## [Decision Letter · Decision Letter 0]

29 Jul 2024

Dear Dr. Ganusov,

Thank you very much for submitting your manuscript "Mathematical modeling suggests heterogeneous replication of Mycobacterium tuberculosis in rabbits" for consideration at PLOS Computational Biology.

As with all papers reviewed by the journal, your manuscript was reviewed by members of the editorial board and by several independent reviewers. In light of the reviews (below this email), we would like to invite the resubmission of a significantly-revised version that takes into account the reviewers' comments.

The reviewers appreciated the paper, and thought it could make a nice addition to the literature. Please address their comments!

We cannot make any decision about publication until we have seen the revised manuscript and your response to the reviewers' comments. Your revised manuscript is also likely to be sent to reviewers for further evaluation.

Sincerely,

Rustom Antia

Academic Editor

PLOS Computational Biology

Amber Smith

Section Editor

PLOS Computational Biology

The reviewers appreciated the paper, and thought it could make a nice addition to the literature. Please address their comments!

Reviewer's Responses to Questions

**Comments to the Authors:**

Reviewer #1: In this paper the authors report results of experiments of Mtb infection of rabbits and modeling of the within rabbit dynamics of the number of bacteria, especially replication and death rates.

Line 71 (and other places): “enters in static equilibrium” or “enters a static equilibrium”?

Line 78: “at such high dose” is perhaps better than “at such higher dose”.

Line 149: It is a little confusing that you say slope of N and P can be used to calculate division and death rates using equations (1) and (2), when these equations don’t include slope of P. It would also be useful to briefly re-state that the form of these equations was derived/presented in Gill et al [17].

Line 153: You say that Mtb is expected to not die in vitro, do you also check for this in the experiments, e.g. Mtb viability? Why do you use the two media, what is the difference?

Line 197: “use the method”

Line 204: This seems to indicate that you expect the fit to be more stable if you use N and f, rather than P and F. Why is that?

Line 305: It is unclear what you mean by negative rates here. Negative r and negative delta? Please explain what that means… This was explained just below, perhaps a call to “see below” would be enough here.

Line 358: Perhaps it is more accurate to say that “Mtb population consists of at lest two sub-populations of bacteria.”, since it is not clear if the model suggests two populations for biological of statistical reasons.

Line 400: You provide the AIC, but not the qui-squared or p-value for the comparison of two vs. three populations.

Line 465: “are likely to be formed by”, the word “be” is missing.

Line 487: This line reads awkwardly, is the word “are” extra?

Line 530: Should this be “about 100x more lung weight” and not 1000x?

The paper is well written overall, and the methods are explicit. Please consider my suggestions above. In addition, a couple of other points: i) what is your estimate of total bacterial replication and death, in absolute numbers, over the ~4 months of the experiment? How many bacteria were produced and killed over this period? ii) a second question, is whether heterogeneity in s (either across bacterial population or over time) could also explain your data? iii) what if s was not independent of replication rate, is this a possibility?

Reviewer #2: Key question: whether variability in granuloma-scale control of bacteria arises due to differences in replication rates or death rate (or both) of bacteria in different lesions remains largely unknown. They suggest it is the balance of death and division rates—and base it nothing on the role of immunity holding things in check. This is surprising and very simplified version of what could be happening.

The premise of the is paper is that “It is well understood that pulmonary TB is due to replication of Mtb in the lung but quantitative details of Mtb replication and death in lungs of patients and how these rates are related to the degree of lung pathology are unknown” and this point is certainly controversial. Many experts believe that the immune response to Mtb bacteria within lungs is what lead to TB disease manifestations so the premise that replication as the sole factor is misinforming.

Other concerns/questions--

Regarding the different kinetic groups of bacteria, it is important to note that bacteria reside in 3 major areas within lung granulomas: 1) trapped with macrophages, extracellular and those that are non-replicating within the caseous necrosis that are still metabolically active but not dividing. There is no discussion of this.

Where and why do these subsets exist?

Continual references to the mouse model are made and all know that the mice is NOT a good model for TB and in fact, they get chronic infection which does not happen in primates or rabbits. There is also a paper on the ultra-low dose mouse model (urdalh • PMCID: PMC7854984 ) that shows that it is closer to what happens in primates than the 100 dose model that the authors refer to.

Line 48,52 chronic stage of infection

—this seems to be only present in mouse models. Reference [26] points out the three relevant animal models for TB

Line 60 isoniazid treatment works but the author did not explain that it takes MONTHS for that to happen…and that gives the bacteria opportunity to change replication rates in that time frame.

Almost all TB scientists believe that Mtb is replicating during LTBI and that it is the immune response that keeps it in check.

Perhaps the reason that the previous model failed to represent the data is that they are ignoring the large piece of immune control? And as stated it is well-accepted that there are multiple subpopulations of Mtb in a host (see any paper on PK/PD of treating TB)

Their authors have not made it clear that while their mathematical model can track different replication rates, it does not explain that OTHER FACTORS are at work influencing the dynamics. For example, these authors are well-acquainted with the relational to a viral dynamics of HIV/HepC and the viral set point has a very similar relationship to the MTb setpoint in TB. (in primates and rabbits, not mice). Thus, the leveling off of bacteria is driven by the immune response, as is the viral replication load in these other infections. How can those features not be included even phenomenologically in their new model?

Line 153--Is the assumption of “In vitro we expect that bacteria do not die (i.e., delta = 0)

Assumption for all time or only the length of the experiment? How useful is this assumption when an immune response is present?

Line 189-“ We propose that a non-monotonous change in the percent of plasmid-bearing cells in the bacterial population may arise due to asynchronous dynamics of several independent su populations of bacteria.”

-What would lead to these different independent populations? Why would they exist?

Line 265: We do not know why the dynamics of bacterial growth changed after 6 days of culture; it may be related to the adaptation of the natural Mtb isolate HN878 to the growth in culture. This dynamics is not consistent with the continuous monotonic increase in the concentration of H37Rv-pBP10 in 7H9 media of different dilutions (e.g., Figure 1d in Gill et al. [17]).

-This is very strange to hang the whole story on this in vitro culture and no real understanding is given for the dynamics?

Line 290- “500 to nearly 108 in the  first 28 days, then the number of

bacteria declined for the next 56 days (until 84 days), and then the number of bacteria exploded to about 1010 CFU/lungs in some rabbits or was maintained at 107 CFU/lungs in others

-this is exactly tied to the “bacterial set point idea” raised above and the timinig is the length of time for the generation of adaptive immunity in rabbits to indice the turn around at day 28. The explosion in bacteria is due to Rabbits being a progressive model of TB and the immune response is unable to control infection.

the authors could not explain the results with the original model of Gill et al so build another two-population model (although it is likely 3 as mentioned above) and show it could work. There is no basis for this 2-sub pop model and again, could a model of immune response and Mtb (similar to the HIV models) explain the same thing?

Line 330-And predicted no Mtb replication between 28 and 56 days post infection.

-which of course there is no evidence for, and it is the immune system that is keeping it in check.

Line 377” Then to explain the increase in the percent of plasmid-bearing cells, while the total number of bacteria declines between 28 and 56 days post infection, one must assume that the initial dominant population contracts which another population which another population with a higher frequency of plasmid-bearing cells (due to fewer divisions) becomes more dominant

—again what is the mechanism or driving biology for this segregation? Perhaps their results of the two-pop sub model could also be explained that some bacteria are being taken up by macrophages and some are being trapped in caseous necrosis? The environments lead to different replication rates as well as any that remain extracellular. Rather than a new assumption

Line 398 Interestingly, while the three population model  fitted the data statistically better than either a single population model

-As expected as 3 different populations are well-documented…and this gives biological plausibility to their argument. Not discussed and should be.

**Have the authors made all data and (if applicable) computational code underlying the findings in their manuscript fully available?**

Reviewer #1: Yes

Reviewer #2: Yes

PLOS authors have the option to publish the peer review history of their article (what does this mean?). If published, this will include your full peer review and any attached files.

Reviewer #1: No

Reviewer #2: No
---

## [Decision Letter · Decision Letter 1]

22 Sep 2024

Dear Dr. Ganusov,

Thank you very much for submitting your manuscript "Mathematical modeling suggests heterogeneous replication of Mycobacterium tuberculosis in rabbits" for consideration at PLOS Computational Biology. As with all papers reviewed by the journal, your manuscript was reviewed by members of the editorial board and by several independent reviewers. The reviewers appreciated the attention to an important topic. Based on the reviews, we are likely to accept this manuscript for publication, providing that you modify the manuscript according to the review recommendations.

Reviewer 2 notes that you have replied extensively to the reviewers comments but only made small edits to the manuscript. I would suggest you can and should do a better job editing the manuscript to address some of the issues rather than only replying to the reviewers. Please do so and I believe the manuscript will be acceptable for publication.

Sincerely,

Rustom Antia

Academic Editor

PLOS Computational Biology

Amber Smith

Section Editor

PLOS Computational Biology

Reviewer 2 notes that you have replied extensively to the reviewers comments but only made small edits to the manuscript. I would suggest you can and should do a better job editing the manuscript to address some of the issues rather than only replying to the reviewers. Please do so and I believe the manuscript will be acceptable for publication.

Reviewer's Responses to Questions

**Comments to the Authors:**

Reviewer #1: Thank you for reviewing the manuscript and carefully answering my previous questions. I think the manuscript has improved and I don't have further questions.

However, a small note. In your response to one of my previous questions, regarding estimating total bacteria replication and death, you calculated this by making delta=0. I am not sure this is the best way. I would use the model simulation with the estimated delta and simply calculate the integral of production and death (for instance with two new accounting compartments for rP and deltaP, etc). This is because without death, rP is going to be larger at every time, which generates higher number of bacteria in a compound effect. I realize that you decided to not add this calculation to your manuscript, so this is a side note.

Reviewer #2: the authors gave lengthy detailed replies to the queries raised, however only very minor edits were actually incorporated into the manuscript. This review believes that some of the same confusions observed may also be observed by other readers therefore the authors should work in a little more of the replies into the ms. Also, can the authors add something to the abstract explaining the relevance and importance of their work, given that they are not considering other factors that could affect bacterial growth rates in vivo such as location, immunity etc. this reviewer is still left wondering the relevance of the work.

**Have the authors made all data and (if applicable) computational code underlying the findings in their manuscript fully available?**

Reviewer #1: Yes

Reviewer #2: Yes

PLOS authors have the option to publish the peer review history of their article (what does this mean?). If published, this will include your full peer review and any attached files.

Reviewer #1: No

Reviewer #2: No

Figure Files:

Data Requirements:

Reproducibility:

References:

---

## [Editor Report · Decision Letter 2]

16 Oct 2024

Dear Dr. Ganusov,

We are pleased to inform you that your manuscript 'Mathematical modeling suggests heterogeneous replication of Mycobacterium tuberculosis in rabbits' has been provisionally accepted for publication in PLOS Computational Biology.

Best regards,

Rustom Antia

Academic Editor

PLOS Computational Biology

Amber Smith

Section Editor

PLOS Computational Biology

---

## [Editor Report · Acceptance letter]

12 Nov 2024

PCOMPBIOL-D-24-00973R2 

Mathematical modeling suggests heterogeneous replication of Mycobacterium tuberculosis in rabbits

Dear Dr Ganusov,

I am pleased to inform you that your manuscript has been formally accepted for publication in PLOS Computational Biology. Your manuscript is now with our production department and you will be notified of the publication date in due course.

With kind regards,

Lilla Horvath
